# Efficacy of drug treatment for severe melioidosis and eradication treatment of melioidosis: A systematic review and network meta-analysis

**Thunyarat Anothaisintawee**[1]*, **Krit Harncharoenkul**[1], **Kamonporn Poramathikul**[1], **Kittijarankon Phontham**[1], **Parat Boonyarangka**[1], **Worachet Kuntawunginn**[1], **Michele Spring**[1,2], **Daniel Boudreaux**[1], **Jeffrey Livezey**[1], **Narisara Chantratita**[3]

1 Department of Bacterial and Parasitic Diseases, US Army Medical Directorate of the Armed Forces Research Institute of Medical Sciences, Bangkok, Thailand, 2 The Henry M. Jackson Foundation for the Advancement of Military Medicine, Inc., Bethesda, Maryland, United States of America, 3 Department of Microbiology and Immunology, Faculty of Tropical Medicine, Mahidol University, Bangkok, Thailand

* thunyarata@afrims.org

## Abstract

### Background

This systematic review and network meta-analysis (NMA) aimed to compare the efficacy of all available treatments for severe melioidosis in decreasing hospital mortality and to identify eradication therapies with low disease recurrence rates and minimal risk of adverse drug events (AEs).

### Methodology

Relevant randomized controlled trials (RCT) were searched from Medline and Scopus databases from their inception until July 31, 2022. RCTs that compared the efficacy between treatment regimens for severe melioidosis or eradication therapy of melioidosis, measured outcomes of in-hospital mortality, disease recurrence, drug discontinuation, or AEs, were included for review. A two-stage NMA with the surface under the cumulative ranking curve (SUCRA) was used to estimate the comparative efficacy of treatment regimens.

### Principal findings

Fourteen RCTs were included in the review. Ceftazidime plus granulocyte colony-stimulating factor (G-CSF), ceftazidime plus trimethoprim-sulfamethoxazole (TMP-SMX), and cefoperazone-sulbactam plus TMP-SMX had a lower mortality rate than other treatments and were ranked as the top three most appropriate treatments for severe melioidosis with the SUCRA of 79.7%, 66.6%, and 55.7%, respectively. However, these results were not statistically significant. For eradication therapy, treatment with doxycycline monotherapy for 20 weeks was associated with a significantly higher risk of disease recurrence than regimens containing TMP-SMX (i.e.,TMP-SMX for 20 weeks, TMP-SMX plus doxycycline plus chloramphenicol for more than 12 weeks, and TMP-SMX plus doxycycline for more than 12

**Data Availability Statement:** All relevant data are within the paper and its Supporting Information files.

**Funding:** This study is funded by the Defense Threat Reduction Agency (https://www.dtra.mil; award number HDTRA1239831 to DB). The funder has no role in the study design, data collection and analysis, decision to publish, or preparation of the manuscript.

**Competing interests:** The authors have declared that no competing interests exist.

weeks). According to the SUCRA, TMP-SMX for 20 weeks was ranked as the most efficacious eradication treatment (87.7%) with the lowest chance of drug discontinuation (86.4%), while TMP-SMX for 12 weeks had the lowest risk of AEs (95.6%).

## Conclusion

Our results found a non-significant benefit of ceftazidime plus G-CSF and ceftazidime plus TMP-SMX over other treatments for severe melioidosis. TMP-SMX for 20 weeks was associated with a lower recurrence rate and minimal risk of adverse drug events compared to other eradication treatments. However, the validity of our NMA may be compromised by the limited number of included studies and discrepancies in certain study parameters. Thus, additional well-designed RCTs are needed to improve the therapy of melioidosis.

## Author summary

Melioidosis is a life-threatening infectious disease with a case fatality rate of 21% in Thailand. Furthermore, among patients who survive the acute disease, approximately 23% experience disease recurrence within one year. Despite the high efficacy of currently recommended mono-antibiotic therapies such as ceftazidime or meropenem, the mortality rate in patients with severe melioidosis remains high, ranging from 6%-37%. Thus, several drugs, such as granulocyte stimulating factor (G-CSF) and trimethoprim-sulfamethoxazole (TMP-SMX), have been added to antibiotic monotherapy to enhance its efficacy. However, the efficacy of combined treatments over monotherapies in decreasing mortality rates remains unclear. Additionally, several regimens are available for eradication therapy, which aim to prevent disease recurrence but their efficacy and potential risk of adverse drug events differ. Thus, we conducted a systematic review and network meta-analysis with the aims of comparing the hospital mortality among all available treatments for severe melioidosis, and identifying eradication therapies that effectively decrease disease recurrence, while also minimizing the risk of adverse drug events. Our findings suggest that ceftazidime plus G-CSF and ceftazidime plus TMP-SMX have lower mortality rates than other medications for treating severe melioidosis. However, this effect did not reach statistical significance. For eradication, TMP-SMX for 20 weeks was associated with a lower recurrence rate and a lower risk of adverse drug events when compared to other regimens. However, the validity of our analyses may be compromised due to the low number of included studies and dissimilarity in some factors among the included studies. These findings will be beneficial for clinicians in selecting the appropriate medications for treating severe melioidosis and preventing disease recurrence. In addition, this study suggests that additional well-designed clinical trials are necessary to improve the treatment of melioidosis.

## Background

Melioidosis is a life-threatening infectious disease caused by a gram-negative bacterium, *Burkholderia (B.) pseudomallei*, which is commonly found in soils. *B. pseudomallei* is on the U.S. select agents list (potentially as a bioterrorism agent), and designated as a biosafety level 3 (BSL-3) pathogen by the U.S. Centers of Disease Control and Prevention (CDC) [1]. Southeast

Asia and Australia are highly endemic for melioidosis, with estimated incidence rates ranging from 12.7 per 100,000 persons/year in Thailand to 19.7 per 100,000 persons per year in Australia [2,3]. The incidence of melioidosis is increasing in tropical regions such as Asia, Africa, Central America, and South America [4,5]. Mortality rates in patients with melioidosis varied from 6% in Australia [6] to 40% in other melioidosis-endemic regions [2,7], depending on the severity of the disease, the timing of diagnosis, the type of antibiotics used, and the quality of supportive care. Although the findings from randomized controlled trials (RCTs) have shown that ceftazidime significantly reduced the risk of death in patients with severe melioidosis when compared to conventional therapy (chloramphenicol puls doxycycline plus trimethoprim-sulfamethoxazole (TMP-SMX)), the mortality rate in these patients remains high (6%-37%) [6,8,9], especially in patients with septicemia [10]. Therefore, other treatment regimens (e.g., ceftazidime plus granulocyte colony stimulating factor (G-CSF), ceftazidime plus TMP-SMX, cefoperazone-sulbactam plusTMP-SMX, imipenem, and intravenous amoxicillin/clavulanate) have been investigated in several RCTs [10–13]. However, none of these studies demonstrated a significant benefit of these treatment regimens over ceftazidime in reducing mortality in severe melioidosis patients. The lack of significant differences might be due to inadequate sample sizes in the studies, which led to insufficient power to distinguish between each of the treatment regimens.

*B. pseudomallei* can persist within phagocytes in the human body, particularly in sealed abscesses, where the bactericidal activity of antibiotics is relatively weak [14]. Disease recurrence may occur if the bacterial population resurges. The incidence of disease recurrence ranges from 5.8% in Australia [6] to 25% in Thailand [15], with most occurring in the first year after recovering from acute disease [16]. Therefore, in addition to treatment for the acute phase, oral eradication therapy following the end of parenteral antibiotics is necessary to prevent disease recurrence. The recommended treatment for eradication therapy of melioidosis has changed over time due to adverse drug events (AEs) associated with the previous recommended regimen (i.e., TMP-SMX plus doxycycline for at least 12 weeks) [17]. In Thailand, TMP-SMX alone for 20 weeks is currently recommended for eradication therapy [18], because it was found equally effective as 20 weeks of TMP-SMX plus doxycycline [19]. However, this long-term treatment also increases the risk of AEs, and decreases patient compliance. In Australia, the shorter treatment regimen of TMP-SMX for 12 weeks has been recommended for eradication therapy based on evidence from an observational study [20]. However, the results from a single small RCT conducted in Thailand suggested that TMP-SMX for 12 weeks was inferior to TMP-SMX for 20 weeks in preventing disease recurrence but not in reducing mortality [21]. Due to these varying conclusions, a more comprehensive comparison of the efficacy of all available treatment regimens is needed to make a recommendation for the eradication treatment of melioidosis.

In 2002, a Cochrane review comprehensively assessed the efficacy of interventions for treating melioidosis [22]. This review concluded that treatment regimens with ceftazidime were significantly more effective in reducing mortality compared to chloramphenicol doxycycline, and TMP-SMX. For oral therapy in the maintenance phase, the death rate of chloramphenicol, doxycycline, and TMP-SMX was not significantly different from other regimens such as amoxycillin-clavulanic acid, ciprofloxacin-azithromycin, and doxycycline alone. However, since the 2002 Cochrane review, six new RCTs [10,12,18,19,21,23] have assessed the efficacy of new treatments that were not included in this review. In addition, the Cochrane review did not perform network meta-analysis (NMA) to compare all available treatments simultaneously in a single analysis.

Network meta-analysis is a statistical method that can combine both direct and indirect evidence across a network of studies. Direct evidence refers to a head-to-head RCT that directly

compares two or more interventions of interest, while indirect evidence refers to a comparison of the interventions via one or more common comparators. Network meta-analysis produces estimates of the relative treatment effects between any pair of interventions in the network and usually yields more precise estimates than a single direct or indirect estimate [24]. In addition, NMA can simultaneously estimate relative efficacy and safety between available drug treatments and estimatethe probability of the best treatment among all available regimens, considering both benefit (e.g., hospital mortality, disease recurrence) and risk (e.g., AEs). Thus, we performed a systematic review and NMA of RCTs aiming to 1) compare the efficacy in decreasing mortality among all available treatments for severe melioidosis, and 2) compare the efficacy in reducing disease recurrence, and the risk of AEs among all available eradication treatments for melioidosis. The results from this study will aid clinicians in selecting the most appropriate treatment for melioidosis when balancing the benefit and risks of each treatment regimen.

## Methods

This systematic review and network meta-analysis was conducted and reported according to the Preferred Reporting Items for Systematic Reviews and Meta-Analyses (PRISMA) extension statement for reporting systematic reviews incorporating network meta-analyses of health care interventions [25]. The review protocol is registered on the PROSPERO website (CRD42022345699).

We searched the relevant RCTs from Medline and Scopus databases from inception through July 31, 2022. The search terms and search strategies for each database are as follows.

### Scopus database

(melioidosis OR "Burkholderia pseudomallei") AND ("clinical trial" OR randomised OR randomized) AND (LIMIT-TO (DOCTYPE,"ar"))

### Medline database

(("Melioidosis"[Mesh]) OR (melioidosis)) OR (Burkholderia pseudomallei); Filters: Clinical Trial, Clinical Trial Protocol, Clinical Trial/Phase I, Clinical Trial/Phase II, Clinical Trial/ Phase III, Clinical Trial/Phase IV, Comparative Study, Controlled Clinical Trial, Pragmatic Clinical Trial, Randomized Controlled Trial

Two reviewers (KP1 and KP2) independently selected the studies based on titles and abstracts. Full articles were reviewed if a decision could not be made based on titles and abstracts. Disagreement between the 2 reviewers was resolved using a the third party (TA) to form a consensus. RCTs were eligible for review if they included patients diagnosed with melioidosis, compared the efficacy between treatment regimens that aimed to treat severe melioidosis or eradication of melioidosis, and measured the outcomes as in-hospital mortality for severe melioidosis, disease recurrence, or any adverse drug events which occurred during eradication therapy.

### Interventions of interest

Interventions of interest were drug regimens used for the treatment of severe melioidosis or eradication treatment of melioidosis. Severe melioidosis was defined according to the criteria reported in each study, such as sepsis, organ dysfunction, or hypotension. Treatment regimens for severe melioidosis refer to intravenous medications provided during the acute phase, including ceftazidime, ceftazidime plus TMP-SMX, imipenem, chloramphenicol plus

doxycycline plus TMP-SMX, amoxicillin/clavulanic acid, cefoperazone-sulbactam plus TMP-SMX, and ceftazidime plus G-CSF. Eradication treatment is defined as an oral therapy provided after the end of initial parenteral therapy to prevent recurrent melioidosis. Eradication therapy consisted of TMP-SMX plus chloramphenicol plus doxycycline for >12 weeks, TMP-SMX plus doxycycline for >12 weeks, TMP-SMX for 20 weeks, TMP-SMX for 12 weeks, doxycycline for 20 weeks, ciprofloxacin plus azithromycin for 12 weeks, and amoxicillin/clavulanic acid for 20 weeks.

## Outcomes of interest

Primary outcomes were hospital mortality for severe melioidosis and disease recurrence for eradication therapy. Disease recurrence consisted of culture-confimed and clinically-suspected recurrence. Culture-confirmed recurrence is defined as new symptoms consistent with melioidosis in association with a positive culture for *B pseudomallei*. Clinically-suspected recurrence is defined as developing new symptoms consistent with melioidosis but in the absence of a positive *B. pseudomallei* culture from any site. Secondary outcomes for eradication treatment were drug discontinuation or switching to other therapies due to treatment failure, serious side effects, and adverse drug events.

## Data extraction

Characteristics of the included studies (i.e., author's name, year of publication, study design, setting), patient's characteristics (i.e., mean age, sex, underlying diseases, disease complications, and laboratory findings at baseline), intervention and comparator's characteristics (i.e., dose and duration of treatment) were independently extracted by two reviewers (TA and KH). The frequency of patients who developed or did not develop outcomes for each treatment was extracted. The corresponding authors were contacted if there was insufficient data.

## Risk of bias assessment

Five domains according to the revised Cochrane risk of bias tool for randomized trials (RoB 2.0) (i.e., bias arising from the randomization process, bias due to deviations from intended interventions, bias due to missing outcome data, bias in the measurement of the outcome, and bias in the selection of reported results) were applied for assessing the methodological qualities of the included studies. The overall risk of bias for each study was classified as low if all the domains for a study were judged as low risk and was classified as high risk of bias if at least one domain for a study was considered as high risk or if multiple domains for a study demonstrated some concerns. Otherwise, the overall risk of bias was classified as some concerns. Two reviewers (TA and KH) independently performed a risk of bias assessment, and disagreements between the two reviewers were resolved by consensus with the third reviewer.

## Data analysis

Only NMA for each outcome was performed because there were not sufficient RCTs that compared similar interventions and outcomes to perform a direct meta-analysis. NMA was performed according to a two-stage frequentist approach. First, risk ratios and variance-covariance for each study were estimated and pooled across studies using a random effect multivariate meta-analysis with a consistency model. The probability of treatments being the best for each outcome was estimated and ranked by a surface under the cumulative ranking curve (SUCRA). The surface under the cumulative ranking curve is a numerical representation of the overall rating that assigns a single value to each treatment. The values of SUCRA range

from 0% to 100%.The greater the SUCRA value and the closer it is to 100 percent, the greater the possibility that a treatment is in the top rank or one of the top rankings; the closer the SUCRA value is to zero, the greater the likelihood that a therapy is in the bottom rank or one of the bottom ranks. A cluster plot of SUCRA values for overall disease recurrence and AEs was constructed to simultaneously assess the benefit and risks of eradication therapy.

The validity of the NMA was assessed by testing the consistency assumption. Inconsistency in the NMA occurs when direct and indirect estimates on a given comparison of interventions disagree [24]. The consistency assumption was assessed using a design-by-treatment model with a global χ2 test [26,27]. If inconsistency was present, the characteristics of studies in the NMA were explored. Small study effects were examined by the comparison-adjusted funnel plots. All analyses were performed using Stata version 17. A two-sided P value $<0.05$ was considered statistically significant for all tests.

## Results

Two-hundred fifty-six (256), 1083, and 10 studies were identified from Medline, Scopus, and ClinicalTrials.gov, respectively. After deleting any duplications, 1328 studies were screened for titles and abstracts. Among them, 14 studies (8 studies for treatment of severe melioidosis and 6 studies for eradication therapy) met the inclusion criteria and were eligible for review (Fig 1). One study of the treatment of severe melioidosis had two sub-trials that compared the exact same treatments [12]. Therefore, when analyzing treatments for severe melioidosis, there were a total of 9 studies included.

### Risk of bias assessment

The risk of bias assessment results are presented in S1 and S2 Figs. Among the 8 included studies for treating severe melioidosis, almost all studies (7/8) were rated as having some concern for the overall domain, and one study had a high risk of bias [8].

The six trials on eradication therapy were classified as having either low (n = 2) [19,21], some concern (n = 2) [18,28], or high risk (n = 2) [29,30] for overall bias. The high risk of bias in the overall domain was due to deviations from the intended interventions and missing outcome data.

### Treatment of severe melioidosis

Characteristics of included studies for the treatment of severe melioidosis are presented in Table 1. All studies were conducted in Thailand. The mean age of the study participants varied from 45 to 59 years, with most of them being males (55%-74%). Diabetes mellitus (DM) and chronic kidney disease (CKD) were found in 22% to 69% and 15% to 42% of the study participants, respectively. Most participants (56% to 79%) had septicemia at baseline, while those with pneumonia at baseline ranged from 49% to 72%. Treatment comparisons varied among nine studies (cefoperazone-sulbactam+TMP-SMX versus (vs.) ceftazidime+TMP-SMX for 2 studies [11,31], ceftazidime+TMP-SMX vs. ceftazidime for 1 study with 2 sub-studies [12], ceftazidime vs. imipenem for 1 study [32], ceftazidime+G-CSF vs. ceftazidime for 1 study [10], chloramphinicol+doxycycline+TMP-SMX vs. ceftazidime+TMP-SMX for 1 study [8], chloramphinicol+doxycycline+TMP-SMX vs. ceftazidime for 1 study [9], and ceftazidime vs. amoxicillin-clavulanate for 1 study [13]). Due to insufficient studies that compared similar interventions and outcomes, a direct meta-analysis could not be performed.

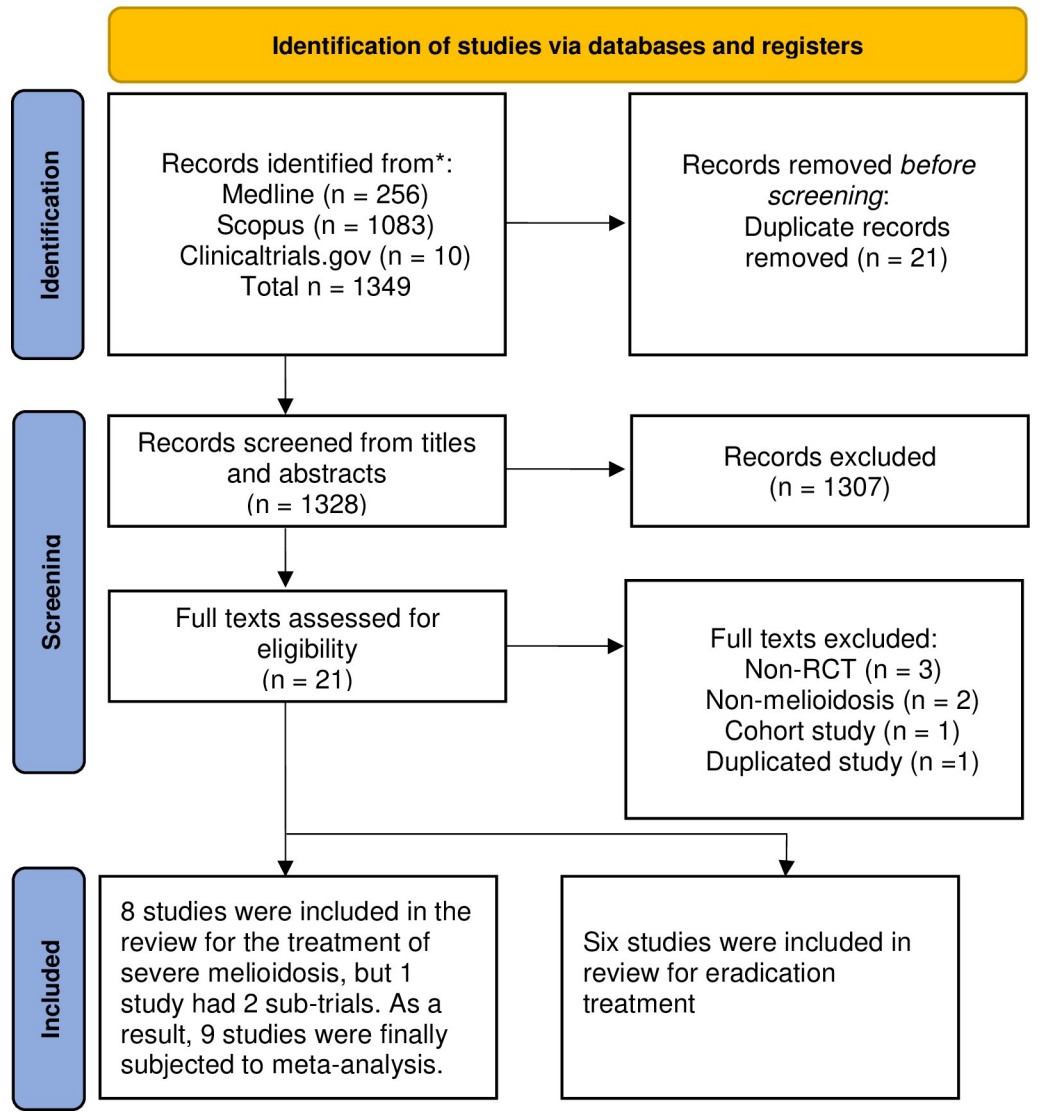

**Fig 1. Flow chart of study selection.**

## Hospital mortality

In a network meta-analysis of hospital mortality, 9 studies covering 7 treatment regimens were considered (S3 Fig and S1 Table). The global test revealed consistent results (Chi$^2$ = 0.15, P = 0.696). Relative treatment effects of all treatment comparisons are presented in Table 2. Treating with chloramphenicol plus doxycycline plus TMP-SMX had a significantly greater risk of hospital mortality than other treatments except for cefoperazone-sulbactam plus TMP-SMX, with RRs of 2.22 (95% CI: 1.28–3.86) for ceftazidime plus TMP-SMX, 1.96 (95% CI: 1.13–3.38) for amoxicillin-clavulanic acid, 2.01 (95% CI: 1.12–3.59) for imipenem, 2.38 (95% CI: 1.39–4.07) for ceftazidime plus G-CSF, and 1.92 (95% CI: 1.21–3.05) for ceftazidime. Ceftazidime plus G-CSF had a lower hospital mortality rate than all other treatments. Moreover, the mortality rate for imipenem was lower than that for ceftazidime (RR = 0.96; 95% CI: 0.67–1.36). This RR, meanwhile, failed to achieve statistical significance. Ceftazidime with TMP-SMX had a lower mortality rate than ceftazidime alone (RR = 0.86; 95% CI: 0.56, 1.34)

**Table 1. Characteristics of included studies for the treatment of severe melioidosis.**

| Author, year | Intervention | Dose | Comparator | Dose | N | Age* | Male (%) | DM (%) | CKD (%) | Septicemia (%) | WBC** | Pneumonia (%) | Cr (mg/dl) | TB (mg/dl) |
|---|---|---|---|---|---|---|---|---|---|---|---|---|---|---|
| Cheng, 2007 [10] | G-CSF + ceftazidime | 263 μg/day | Ceftazidime | NR | 60 | 59 | NR | 55 | 22 | NR | 12700 | 60 | 3.8 | 1.0 |
| Chetchotisakd, 2001 [11] | Cefoperazone-sulbactam + TMP/SMX | 3 g/day, 25 mg/kg + 80:400 8mg/kg/day | Ceftazidime + TMP/SMX | 100 mg/kg/day + 80/400 8 mg/kg/day | 100 | 50 | 68 | 69 | 35 | 59 | NR | 49 | 0.42 | 2.1 |
| Chierakul, 2005 [12] | Ceftazidime + TMP/SMX | 120 mg/kg/day + 160/800 mg every 8 hrs | Ceftazidime | 120 mg/kg/day | 154 | 50 | 56.5 | 65.6 | 16.9 | 55.8 | 13446 | 61.7 | 1.4 | 1.1 |
| Chierakul, 2005 [12] | Ceftazidime + TMP/SMX | 120 mg/kg/day + 160/800 mg every 8 hrs | Ceftazidime | 120 mg/kg/day | 87 | 51 | 73.6 | 57.5 | 14.9 | 59.8 | 12912 | 52.9 | 1.2 | 1.5 |
| Simpson, 1999 [32] | Imipenem | 50 mg/kg/day | Ceftazidime | 120 mg/kg/day | 214 | 51.5 | 55.1 | 48.6 | 19.6 | 61.7 | 12526 | 55.6 | 1.9 | 1.2 |
| Sookpranee, 1992 [8] | Ceftazidime + TMP/SMX | 100 mg/kg/day + 8 mg/kg/day for TMP, 40 mg/kg/day for SMX | Chloramphinicol + doxycycline + TMP/SMX | 100 mg/kg/day +4 mg/kg/day +8 mg/kg/day for TMP, 40 mg/kg/day for SMX | 61 | 44.9 | 64.2 | 57.4 | 24.6 | 68.9 | 70032 | NR | 3.6 | NR |
| Suputtamongkol, 1994 [13] | Ceftazidime | 120 mg/kg/day | Amoxicillin/clavulanate | 160 mg/kg/day | 212 | 48.5 | 61.1 | NR | NR | 67 | 13742 | 64.2 | 1.9 | 2.3 |
| Thamprajamchit, 1998 [31] | Cefoperazone-sulbactam + TMP/SMX | 25 mg/kg/day+ 8 mg/kg/day | Ceftazidime +TMP/SMX | 100 mg/kg/day + 8 mg/kg/day | 38 | 50 | 73.7 | 31.6 | 42.1 | 78.9 | NR | NR | 0.62 | 2.31 |
| White, 1989 [9] | Ceftazidime | 120 mg/kg/day | Chloramphinicol + doxycycline + TMP/SMX | 100 mg/kg/day + 4 mg/kg/day + 10 mg/kg/day for TMP, 50 mg/kg/day for SMX | 65 | 50.6 | 68 | 21.5 | 32.3 | NR | NR | 72.3 | NR | NR |

CKD, chronic kidney disease; Cr, serum creatinine; DM, diabetes mellitus; G-CSF, granulocyte colony-stimulating factor; NR, not reported; TB, total bilirubin; TMP-SMX, trimethoprim-sulfamethoxazole; WBC, white blood cell, *mean, **per microliter

**Table 2. Estimation of relative treatment effects on hospital mortality.** Results are risk ratios (95% confidence intervals) between each pair of treatments from network meta-analysis. Comparisons are read from right to left. For example, the risk ratio for hospital mortality with Ceftazidime+TMP-SMX compared with TMP-SMX + Doxycycline+Chloramphenicol is 0.45 (95% confidence interval: 0.26 to 0.78).

| | | | | | | |
|---|---|---|---|---|---|---|
| Ceftazidime | 0.81 (0.61,1.06) | 0.96 (0.67,1.36) | 0.98 (0.74,1.31) | 0.94 (0.39,2.25) | **1.92 (1.21,3.05)** | 0.86 (0.56,1.34) |
| | Ceftazidime +G-CSF | 1.18 (0.76,1.85) | 1.21 (0.82,1.80) | 1.16 (0.47,2.90) | **2.38 (1.39,4.07)** | 1.07 (0.64,1.79) |
| | | Imipenem | 1.02 (0.65,1.61) | 0.98 (0.38,2.51) | **2.01 (1.12,3.59)** | 0.90 (0.51,1.58) |
| | | | Co-Amoxyclav | 0.96 (0.38,2.40) | **1.96 (1.13,3.38)** | 0.88 (0.52,1.49) |
| | | | | CPZ-SBT+TMP-SMX | 2.04 (0.80,5.20) | 0.92 (0.43,1.95) |
| | | | | | TMP-SMX + Doxycycline +Chloramphenicol | **0.45 (0.26,0.78)** |
| | | | | | | Ceftazidime+TMP-SMX |

The results that are highlighted in blue were exclusively generated from direct estimates. The results produced from both direct and indirect estimations are highlighted in yellow. Only indirect estimations were used to determine the results when indicated in green.

G-CSF, granulocyte colony-stimulating factor; Co-Amoxyclav, amoxicillin/clavulanic acid; CPZ-SBT, Cefoperazone/Sulbactam; TMP-SMX, trimethoprim-sulfamethoxazole

Bold font indicates statistical significance.

and imipenem alone (RR = 0.90; 95% CI: 0.51–1.58). However, none of these RRs achieved statistical significance.

The top three most effective treatments for the outcome of hospital mortality, according to SUCRAs, are ceftazidime plus G-CSF (79.7%, mean rank = 2.2), ceftazidime plus TMP-SMX (66.6%, mean rank = 3.0), and cefoperazone-sulbactam plus TMP-SMX (55.7%, mean rank = 3.7) (S4 Fig). Chloramphenicol plus doxycycline plus TMP-SMX and ceftazidime alone had the lowest SUCRA levels, at 1.7% and 42.8%, respectively. The comparison-adjusted funnel plot was symmetrical, indicating no small-study effects (S5 Fig).

## Eradication therapy

Table 3 presents the characteristics of the six included studies on the eradication therapy of melioidosis. All studies were conducted in Thailand. The participants' average age ranged from 45 to 54 years, and most were male (57%-67%). Participants' DM and CKD rates ranged from 38% to 71% and 4% to 19%, respectively. Around 20% to 43% of participants had bacteremia at baseline, while those with disseminated disease (i.e., a positive blood culture result plus >1 noncontiguous focus of infection) ranged between 14% and 16%. The studies included in the analysis had varying treatment comparisons and treatment durations, which were too different to conduct a direct meta-analysis (see Table 3).

## Disease recurrence

In a network meta-analysis of disease recurrence, six studies involving seven treatment regimens (TMP-SMX plus doxycycline for > 12 weeks, TMP-SMX for 20 weeks, TMP-SMX for 12 weeks, doxycycline for 20 weeks, TMP-SMX plus doxycycline plus chloramphenicol for > 12 weeks, ciprofloxacin plus azithromycin for 12 weeks, and amoxicillin-clavulanic acid for 20 weeks) were evaluated (S6A Fig and S2 Table). The global test indicated consistent results of NMA (Chi$^2$ = 2.36, P = 0.125). Table 4 presents the relative effects of all treatment comparisons. Treatment with doxycycline for 20 weeks was associated with a significantly higher risk of disease recurrence than treatments with TMP-SMX for 20 weeks, TMP-SMX plus doxycycline plus chloramphenicol for > 12 weeks, and TMP-SMX plus doxycycline for > 12 weeks with RRs of 3.54 (95% CI: 1.21,10.37), 2.56 (95% CI: 1.27,5.17), and 3.22 (95%

Table 3. Characteristics of the included studies for eradication therapy.

| Author, year | Intervention | Dose | Duration (week) | Comparator | Dose | Duration (week) | N | Age (mean) | Male (%) | DM (%) | CKD (%) | Bacteremia (%) | Disseminated disease* (%) | Duration of IV RX (week) |
|---|---|---|---|---|---|---|---|---|---|---|---|---|---|---|
| Anunnatsiri, 2020 [21] | TMP/SMX | 160/800 mg BID for BW <40 kg, or 240/1200 mg BID for BW 40–60 kg, or 320/1600 mg BID for BW > 60 kg BID | 12 | TMP/SMX | 160/800 mg BID for BW <40 kg, or 240/1200 mg BID for BW 40–60 kg, or 320/1600 mg BID for BW > 60 kg BID | 20 | 658 | 54 | 67 | 70.5 | 13.4 | 43.1 | 15.5 | NR |
| Chaowagul, 2005 [18] | TMP/SMX + doxycycline + chloramphenicol | 160/800 mg BID + 4 mg/ kg/day + 40 mg/kg/day | > 12 | TMP/SMX + doxycycline | 160/800 mg BID + 4 mg/kg/ day | > 12 | 180 | 47 | 61.7 | 41.7 | 3.9 | 22.2 | 22.8 | 13 |
| Chaowagul, 1999 [29] | TMP/SMX + doxycycline + chloramphenicol | 8/40 mg /kg/ day+ 4 mg/ kg/day + 40 mg/kg/day | 20 | Doxycycline | 4 mg/kg/day | 20 | 109 | 51 | 56.9 | 57.8 | 5.6 | 20.2 | 22 | 16 |
| Chetchotisakd, 2001 [28] | Ciprofloxacin + azithromycin | 20 mg/kg/ day + 500 mg/day | 12 | TMP/SMX + doxycycline | 10 mg/kg/day for TMP, 50 mg/kg/day for SMX + 4 mg/ kg/day | 20 | 65 | 50 | 63.1 | 66.2 | 18.5 | 43.1 | 36.9 | 14 |
| Chetchotisakd, 2014 [19] | TMP/SMX | 160/800 mg BID for BW <40 kg, or 240/1200 mg BID for BW 40–60 kg, or 320/1600 mg BID for BW > 60 kg BID | 20 | TMP/SMX + doxycycline | 160/800 mg BID for BW <40 kg, or 240/1200 mg BID for BW 40–60 kg, or 320/1600 mg BID for BW > 60 kg BID + 100 mg BID | 20 | 626 | 50.5 | 62.5 | 66 | 4.5 | 30 | 14 | NR |
| Rajchanuvong, 1995 [30] | TMP/SMX + doxycycline + chloramphenicol | 10 mg/kg/ day for TMP, 50 mg/kg/ day for SMX + 4 mg/kg/ day + 40 mg/ kg/day | 20 | Co-amoxiclav | 30 mg/kg/day for amoxicillin, 15 mg/kg/day for clavulanic acid | 20 | 101 | 44.6 | 59.4 | 37.6 | 9.9 | NR | NR | NR |

BID, twice daily; BW body weight; Co-amoxiclav, amoxicillin/clavulanic acid; CKD, chronic kidney disease; DM, diabetes mellitus; IV, intravenous; NR, not reported; RX, treatment

*Disseminated disease was defined as a positive blood culture result plus >1 noncontiguous focus of infection.

**Table 4. Estimation of relative treatment effects on disease recurrence (above diagonal line) and drug discontinuation (below diagonal line).** Results are risk ratios (95% confidence intervals) between each pair of treatments from network meta-analysis. Comparisons are read from right to left. For example, the risk ratio for disease recurrence with TMP-SMX 20 weeks compared with TMP-SMX 12 weeks is 0.52 (95% confidence interval: 0.21 to 1.28).

| Disease recurrence | | | | | | |
|---|---|---|---|---|---|---|
| **TMP-SMX + Doxycycline >12 weeks** | 3.00 (0.84,10.79) | 3.09 (0.92,10.40) | 1.26 (0.67,2.37) | **3.22 (1.25,8.30)** | 1.76 (0.62,4.97) | 0.91 (0.55,1.51) |
| **2.36 (1.31,4.25)** | **Co-Amoxiclav 20 weeks** | 1.03 (0.18,6.01) | 0.42 (0.14,1.27) | 1.07 (0.29,3.99) | 0.59 (0.11,3.04) | 0.30 (0.08,1.20) |
| 0.88 (0.33,2.35) | 0.37 (0.12,1.17) | **Ciprofloxacin + Azithromycin 12 weeks** | 0.41 (0.10,1.60) | 1.04 (0.22,4.85) | 0.57 (0.12,2.81) | 0.29 (0.08,1.09) |
| **2.13 (1.41,3.22)** | 0.90 (0.59,1.37) | 2.41 (0.84,6.96) | **TMP-SMX + Doxycycline +Chloramphenicol >12 weeks** | **2.56 (1.27,5.17)** | 1.40 (0.41,4.72) | 0.72 (0.32,1.63) |
| **7.64 (2.52,23.14)** | **3.23 (1.07,9.82)** | **8.65 (1.97,37.84)** | **3.58 (1.28,10.02)** | **Doxycycline 20 weeks** | 0.55 (0.13,2.23) | **0.28 (0.10,0.83)** |
| 0.70 (0.01,35.69) | 0.30 (0.01,15.77) | 0.79 (0.01,45.49) | 0.33 (0.01,17.09) | 0.09 (0.00,5.44) | **TMP-SMX 12 weeks** | 0.52 (0.21,1.28) |
| **0.67 (0.47,0.95)** | **0.28 (0.14,0.56)** | 0.76 (0.27,2.14) | **0.31 (0.18,0.54)** | **0.09 (0.03,0.28)** | 0.96 (0.02,48.16) | **TMP-SMX 20 weeks** |
| Drug discontinuation | | | | | | |

The results that are highlighted in blue were exclusively generated from direct estimates. The results produced from both direct and indirect estimations are highlighted in yellow. Only indirect estimations were used to determine the results when indicated in green.

Co-Amoxyclav, amoxicillin/clavulanic acid; TMP-SMX, trimethoprim-sulfamethoxazole

Bold font indicates statistical significance.

CI: 1.25,8.30), respectively. Treatment with TMP-SMX for 20 weeks had a lower disease recurrence than other therapies, although the results were not statistically significant. According to SUCRA, TMP-SMX for 20 weeks (87.7%, mean rank = 1.7), TMP-SMX plus doxycycline for >12 weeks (81.0%, mean rank = 2.1), and TMP-SMX plus doxycycline plus chloramphenicol for > 12 weeks (67.0%, mean rank = 3.0) have the highest probability to be the most effective medications for reducing melioidosis recurrence (S7 Fig). Treatment with doxycycline for 20 weeks had the lowest SUCRA levels (19.4, mean rank = 5.8). The symmetrical comparison-adjusted funnel plot indicates no small-study effects (S8 Fig).

## Drug discontinuation

A network meta-analysis of drug discontinuation included 6 studies encompassing 7 treatment regimens (i.e., TMP-SMX plus doxycycline for >12 weeks, TMP-SMX for 20 weeks, TMP-SMX for 12 weeks, doxycycline for 20 weeks, TMP-SMX plus doxycycline plus chloramphenicol for > 12 weeks, ciprofloxacin plus azithromycin for 12 weeks, and amoxicillin-clavulanic acid for 20 weeks), see S6B Fig and S3 Table. The global test suggested consistent results (Chi$^2$ = 0.23, P = 0.633). Treatment with doxycycline for 20 weeks had a significantly higher risk of drug discontinuation than other treatments except TMP-SMX for 12 weeks, see Table 4. TMP-SMX plus doxycycline plus chloramphenicol for >12 weeks and amoxicillin-clavulanic acid for 20 weeks were associated with significantly higher drug discontinuation rates than TMP-SMX plus doxycycline for >12 weeks with RRs of 2.13 (95% CI: 1.41, 3.22) and 2.36

(95% CI: 1.31, 4.25), respectively. While, TMP-SMX for 20 weeks had a significantly lower drug discontinuation rate than doxycycline for 20 weeks, TMP-SMX plus doxycycline for >12 weeks, TMP-SMX plus doxycycline plus chloramphenicol for >12 weeks, and amoxicillin-cla-vulanic acid for 20 weeks. According to SUCRAs, the therapy with the lowest probability of drug discontinuation is TMP-SMX for 20 weeks (86.4%, mean rank = 1.8), followed by cipro-floxacin plus azithromycin for 12 weeks (71.2%, mean rank = 2.7). The lowest SUCRA levels with the highest probability of drug discontinuation were seen in doxycycline for 20 weeks and amoxicillin-clavulanic acid for 20 weeks at 1.7% and 42.8%, respectively (S7 Fig). The compari-son-adjusted funnel plot was symmetric, indicating no small-study effects existed (S8 Fig).

## Adverse drug events

Initially, the NMA of AEs in 6 studies revealed inconsistency from the global test (Chi$^2$ = 4.06, P = 0.044). After exploring the characteristics of these 6 studies, the study of Rajchanuvong et al. [30] was excluded because of having a lower mean age and a lower percentage of DM than other studies. The network consistency improved after removing this study (Chi$^2$ = 1.28, P = 0.257). Five studies involving six treatment regimens (i.e., TMP-SMX plus doxycycline for > 12 weeks, TMP-SMX for 20 weeks, TMP-SMX for 12 weeks, doxycycline for 20 weeks, TMP-SMX plus doxycycline plus chloramphenicol for >12 weeks, and ciprofloxacin plus azi-thromycin for 12 weeks) were finally included in the NMA of adverse drug events (S6C Fig and S4 Table). The relative effects of all treatment comparisons are shown in Table 5. TMP-SMX for 12 weeks had significantly lower risk of AEs than all other regimens except cip-rofloxacin plus azithromycin with RRs of 0.55 (95% CI: 0.34, 0.88) for TMP-SMX for 20 weeks, 0.40 (95% CI: 0.24, 0.67) for TMP-SMX plus doxycycline for >12 weeks, 0.22 (95% CI: 0.11, 0.44) for TMP-SMX plus doxycycline plus chloramphenicol for >12 weeks, and 0.34

**Table 5. Estimation of relative treatment effects on adverse drug events.** Results are risk ratios (95% confidence intervals) between each pair of treatments from network meta-analysis. Comparisons are read from right to left. For example, the risk ratio for adverse drug events with TMP-SMX 20 weeks compared with TMP-SMX 12 weeks is 1.83 (95% confidence interval: 1.14 to 2.96).

| TMP-SMX + Doxycycline >12 weeks | 0.77 (0.19,3.19) | **1.81 (1.14,2.86)** | 1.19 (0.50,2.81) | **0.40 (0.24,0.67)** | **0.74 (0.62,0.88)** |
|---|---|---|---|---|---|
| | Ciprofloxacin + Azithromycin 12 weeks | 2.34 (0.53,10.36) | 1.54 (0.29,8.05) | 0.52 (0.12,2.35) | 0.96 (0.23,3.98) |
| | | TMP-SMX + Doxycycline + Chloramphenicol >12 weeks | 0.66 (0.32,1.36) | **0.22 (0.11,0.44)** | **0.41 (0.25,0.67)** |
| | | | Doxycycline 20 weeks | **0.34 (0.13,0.92)** | 0.62 (0.26,1.49) |
| | | | | TMP-SMX 12 weeks | **1.83 (1.14,2.96)** |
| | | | | | TMP-SMX 20 weeks |

The results that are highlighted in blue were exclusively generated from direct estimates. The results produced from both direct and indirect estimations are highlighted in yellow. Only indirect estimations were used to determine the results when indicated in green.

TMP-SMX, trimethoprim-sulfamethoxazole

Bold font indicates statistical significance.

(95% CI: 0.13, 0.92) for doxycycline for 20 weeks. Moreover, TMP-SMX treatment for 20 weeks resulted in a reduced risk of adverse drug events compared to treating with TMP-SMX plus doxycycline for >12 weeks and TMP-SMX plus doxycycline plus chloramphenicol for >12 weeks with RRs of 0.74 (95% CI: 0.62, 0.88) and 0.41 (95% CI: 0.25, 0.67). In addition, TMP-SMX plus doxycycline plus chloramphenicol for >12 weeks had a significantly higher risk of adverse drug events than TMP-SMX plus doxycycline for >12 weeks with a RR of 1.81 (95% CI: 1.14, 2.86).

According to SUCRA, TMP-SMX for 12 weeks (95.6%, mean rank = 1.2) was the most effective treatment for minimizing adverse drug events, followed by TMP-SMX for 20 weeks (67.5%, mean rank = 2.6). Alternatively, treatment with TMP-SMX plus doxycycline plus chloramphenicol for >12 weeks was associated with the lowest SUCRA levels (4.9%, mean rank = 5.8), see S9 Fig. The comparison-adjusted funnel plot was symmetric, indicating no small study effects existed (S10 Fig).

## Cluster ranking plot for eradication therapy

The SUCRA clustered ranking plot for disease recurrence, and AEs revealed that TMP-SMX for 20 weeks is the most effective treatment (87.7%) for eradication therapy of melioidosis with a low risk of adverse drug events (67.5%) (Fig 2). TMP-SMX plus doxycycline for >12 weeks also demonstrated efficacy in preventing disease recurrence but had a high risk of adverse drug events. TMP-SMX for 12 weeks, in contrast, demonstrated low efficacy in preventing disease recurrence despite its high treatment acceptability. Doxycycline for 20 weeks had the lowest efficacy and highest risk of adverse drug events.

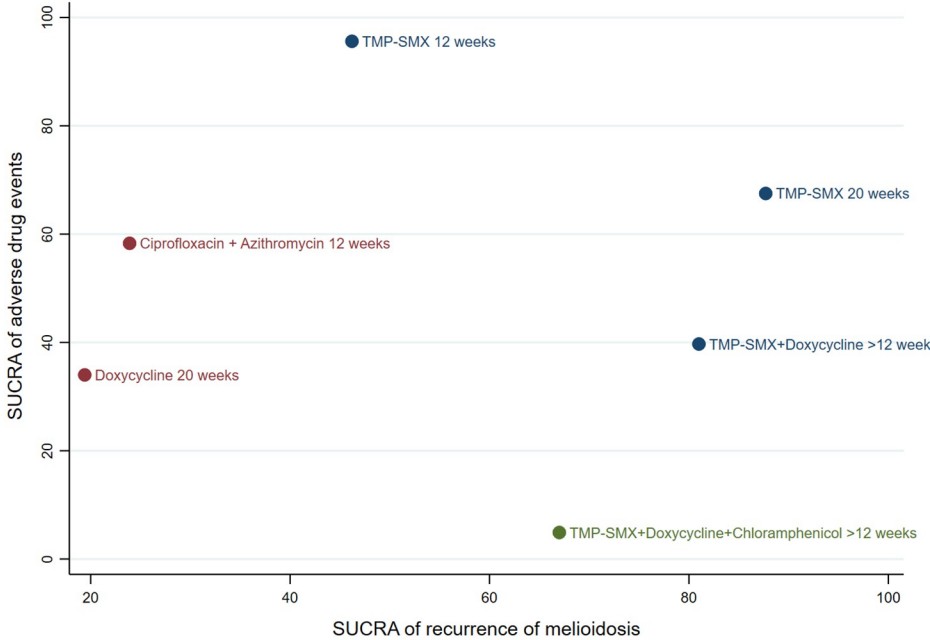

**Fig 2. Cluster ranking plot of surface under cumulative ranking curves (SUCRA) of disease recurrence and adverse drug events for eradication therapy of melioidosis.** The plot is based on cluster analysis of SUCRA values. Each plot represents SUCRA values for two outcomes (i.e., disease recurrence and adverse drug events). Treatments in the upper right corner are more effective (i.e., decreased recurrent rate) and safer (i.e., lower risk of adverse events) compared with other eradication treatments.

## Discussion

Our network meta-analysis found that chloramphenicol plus doxycycline plus TMP-SMX had a significantly higher risk of hospital mortality than other treatments. In contrast, ceftazidime plus G-CSF and ceftazidime plus TMP-SMX had lower hospital mortality rates than other medications. However, the benefit of these two treatments did not reach statistical significance. Furthermore, TMP-SMX for 20 weeks was rated as the most effective eradication treatment for preventing recurrent melioidosis with the lowest possibility of drug discontinuation, while TMP-SMX for 12 weeks was rated as having the lowest risk of adverse drug events. When considering both benefits (prevention of recurrent infection) and risk (adverse drug events), TMP-SMX for 20 weeks seems to be the most appropriate treatment for eradication therapy of melioidosis.

Melioidosis is a severe infectious disease with a high mortality rate. Currently, ceftazidime, meropenem, or imipenem are the preferred first-line treatments for severe melioidosis according to the treatment guidelines from Thailand and Australia [33,34]. However, the mortality rate in patients receiving ceftazidime is still high. Thus, a new treatment regimen is needed to reduce the risk of death in patients with severe melioidosis. Our results suggest that ceftazidime plus G-CSF might have a lower hospital mortality rate than other treatments for severe melioidosis. G-CSF is a hematopoietic growth factor that promotes neutrophil function and is commonly used in patients with neutropenia secondary to chemotherapy-induced myelosuppression. G-CSF also has immunomodulatory effects that can improve the immune functions in patients with sepsis [35]. The findings from a cohort study conducted in Australia reveal that adding G-CSF to antibiotic treatments significantly decreased the risk of death in severe melioidosis patients, compared to conventional treatments [36]. However, evidence from a RCT conducted in Thailand did not find a significant benefit of ceftazidime plus G-CSF over ceftazidime alone, even though the mortality rates between these two regimens were clinically different (83% vs 96%) [10]. These findings are consistent with our results that ceftazidime plus G-CSF had a lower mortality rate than all other treatments and was ranked as the most effective treatment for severe melioidosis. However, evidence from more recent RCTs [37,38] and meta-analysis [39] indicate that G-CSF had no significant benefit in reducing mortality rates in patients with sepsis from other diseases. Nevertheless, our study did not yield statistically significant results, possibly due to an inadequate sample size, which resulted in low power to detect a difference. Therefore, it is important to interpret our findings cautiously. To confirm the effectiveness of G-CSF in patients with severe melioidosis, further multi-center RCTs with larger sample sizes are needed. Moreover, previous RCTs assessing the efficacy of G-CSF included only patients with septicemia. Thus, the benefit of G-CSF may be more significant in patients with lower disease severity. Further studies that include patients with both sepsis and non-sepsis and measure other outcomes, such as acute renal failure or pneumonia, are necessary to confirm this hypothesis.

According to the SUCRA values, the combination of ceftazidime and TMP-SMX is the second most effective treatment for severe melioidosis. However, this result was not statistically significant and the use of TMP-SMX may result in significant adverse drug events. As a result, TMP-SMX should not be routinely prescribed for patients with severe melioidosis. According to the treatment guideline developed by the Darwin group, TMP-SMX is only recommended during the intensive phase of therapy for patients with cutaneous melioidosis, osteomyelitis, septic arthritis, central nervous system infection, or deep-seated collections [40].

When comparing ceftazidime with imipenem, the SUCRA of imipenem for hospital mortality was higher than that of ceftazidime, but the mortality rate was not significantly different between these two drugs. An *in vitro* study found that the bactericidal activity of ceftazidime

was extremely slow, whereas imipenem was bactericidal (99.9% killing rate) after four hours [41]. However, our analysis indicated that imipenem did not provide a better overall prognosis compared to ceftazidime. The results from an RCT conducted in Thailand showed that patients who first received ceftazidime had considerably higher treatment failures (patients who died within 48 hours or required a switch to imipenem) than imipenem [32].

Evidence for meropenem, another drug in the carbapenem class, is only available from observational studies because an RCT comparing ceftazidime with meropenem was withdrawn due to a lack of support from the pharmaceutical company [42]. The Darwin Prospective Melioidosis Study, an observational study that collected data from 1148 melioidosis patients, found that the mortality rate in patients with melioidosis decreased from 31% during 1989–1994 to 6% during 2015–2019 [6,43]. The reduction in mortality rate over time observed in this study might be due to the recommendation in 1998 to use meropenem as the first-line antibiotic for melioidosis, [40] which has been prescribed to 90% of melioidosis patients admitted to the intensive care unit in Australia [43]. In contrast to an RCT, which would have difficulty accounting for the substantial variability across patients in terms of the sites and extent of infection, observational studies using real world data are able to do so. Thus, meropenem, or perhaps meropenem in combination with another drug, maybe a promising choice for the treatment of severe melioidosis.

Our findings suggest that TMP-SMX for 20 weeks should be the most appropriate treatment for eradication therapy when considering both benefits (prevention of recurrent disease) and risk (adverse drug events). This finding aligns with the current practice in Thailand that TMP-SMX for 20 weeks is used as a standard treatment of eradication therapy of melioidosis [19]. However, this regimen has a long duration of treatment. Hence, the patients are prone to adverse drug events and are susceptible to drug discontinuation. In Australia, the TMP-SMX for 12 weeks is recommended as the standard eradication therapy based on evidence from a cohort study showing a low rate of recurrent melioidosis among patients receiving this regimen [44]. However, a more recent RCT conducted in Thailand contradicted this by showing that TMP-SMX for 12 weeks was inferior to TMP-SMX for 20 weeks to prevent disease recurrence [21]. The results from this RCT correspond with our findings. Although TMP-SMX for 20 weeks had a significantly higher risk of adverse drug events than TMP-SMX for 12 weeks. It decreased the chance of disease recurrence by 50%compared to TMP-SMX for 12 weeks. In addition, according to SUCRA values, the risk of drug discontinuation of TMP-SMX for 20 weeks was lower than TMP-SMX for 12 weeks. This might be due to a higher efficacy of TMP-SMX for 20 weeks in preventing disease recurrence, subsequently decreasing the likelihood of switching to other treatments. However, the data of drug discontinuation in TMP-SMX for 12 weeks was based on the results from only one RCT and might be subject to the uncertainty of evidence. Additional studies that compare the efficacy between TMP-SMX for 12 weeks and TMP-SMX for 20 weeks are required to confirm our study's results.

## Strengths and limitations

Our study is the first systematic review and NMA to compare the efficacy of all available treatment regimens for eradication therapy and severe melioidosis treatment. The SUCRA and cluster ranking plots incorporating both benefit (prevention of disease recurrence) and risk (adverse drug events) outcomes were used to identify the best treatment regimen. Consequently, our findings represent treatments with a low recurrent rate and a low risk of adverse drug events.

However, some limitations could not be avoided from our study. First, the number of studies with similar treatment comparisons and outcomes is low preventing us from performing a direct meta-analysis. Second, due to the low number of included studies, most of the findings

from NMA are imprecise and do not achieve statistical significance. Thus, additional head-to-head RCTs that assess the efficacy of different regimens of melioidosis are needed to confirm the results of our study. Also, the majority of studies investigating the treatment of severe melioidosis had some concerns regarding the overall risk of bias. In particular, most studies focusing on eradication therapy were classified as having a high risk and some concerns for the overall risk of bias assessment.

Lastly, while a network meta-analysis has the advantage of being able to compare multiple interventions simultaneously in a single analysis and estimate the relative treatment effects between any pair of interventions in the network, it relies on the comparability of characteristics other than the intervention being compared among the studies included in the network. In other words, for an NMA to be valid, the studies included must be similar in important ways beyond just the treatment being investigated. Studies included in the NMA may differ in various clinical (e.g., age, underlying diseases, and disease severity in study participants) and methodological (e.g., duration of treatment, co-interventions) aspects. These clinical and methodological factors can impact the intervention's effectiveness. As a result, the NMA's validity could be compromised by combining data from studies with distinct clinical and methodological characteristics.

In our NMA, participant characteristics such as mean age, percentage of males, and percentage of septicemia, were generally similar across studies. However, there were discrepancies in the study periods, which ranged from 1989 to 2020. It is worth noting that supportive treatment in critical care units, which has contributed to the decrease in mortality rates, is likely time-dependent. Therefore, differences in study periods could potentially impact the validity of our analysis and result in misleading conclusions. In addition, all included studies were conducted in Thailand, potentially limiting the generalizability of the results in the other settings due to the differences in supportive treatments, the rapidity of disease diagnosis, and organism strains from other countries.

## Conclusion

According to our findings, the combination of chloramphenicol, doxycycline, and TMP-SMX had the highest risk of hospital mortality, while ceftazidime in combination with G-CSF and ceftazidime in combination with TMP-SMX had comparatively lower mortality rates than other medications. However, it is importatnt to note that the benefit of these two therapies did not achieve statistical significance. In terms of eradication therapy, TMP-SMX for 20 weeks was found to be associated with a lower recurrence rate and a decreased risk of adverse drug events compared to other eradication treatments. Nevertheless, due to the small number of studies included and differences in some study parameters, the validity of our NMA may be jeopardized. Therefore, additional well-designed clinical trials are needed to improve the treatment of melioidosis.

## Supporting information

**S1 Table. Treatment comparisons and data used for network meta-analysis of severe melioidosis.**
(DOCX)

**S2 Table. Treatment comparisons and data used for network meta-analysis of eradication therapy (disease recurrence).**
(DOCX)

**S3 Table. Treatment comparisons and data used for network meta-analysis of eradication therapy (drug discontinuation).**
(DOCX)

**S4 Table. Treatment comparisons and data used for network meta-analysis of eradication therapy (adverse drug events).**
(DOCX)

**S1 Fig. Risk of bias assessment for treatment of severe melioidosis.**
(DOCX)

**S2 Fig. Risk of bias assessment for eradication therapy of melioidosis.**
(DOCX)

**S3 Fig. Network map of hospital mortality for treatment of severe melioidosis.**
(DOCX)

**S4 Fig. Surface under the cumulative ranking curve of hospital mortality for treatment of severe melioidosis.**
(DOCX)

**S5 Fig. Comparison adjusted funnel plot of hospital mortality for treatment of severe melioidosis.**
(DOCX)

**S6 Fig. Network map of the outcomes for eradication therapy.**
(DOCX)

**S7 Fig. Surface under the cumulative ranking curve of disease recurrence.**
(DOCX)

**S8 Fig. Comparison adjusted funnel plot of disease recurrence.**
(DOCX)

**S9 Fig. Surface under the cumulative ranking curve of drug discontinuation.**
(DOCX)

**S10 Fig. Comparison adjusted funnel plot for the outcome of drug discontinuation.**
(DOCX)

**S11 Fig. Surface under the cumulative ranking curve of adverse drug events.**
(DOCX)

**S12 Fig. Comparison adjusted funnel plot of adverse drug events.**
(DOCX)

**S1 PRISMA Checklist. PRISMA NMA Checklist.**
(PDF)

## Author Contributions

**Conceptualization:** Thunyarat Anothaisintawee, Worachet Kuntawunginn, Michele Spring, Daniel Boudreaux, Jeffrey Livezey, Narisara Chantratita.

**Data curation:** Thunyarat Anothaisintawee, Krit Harncharoenkul, Kamonporn Poramathikul, Kittijarankon Phontham, Parat Boonyarangka.

**Formal analysis:** Thunyarat Anothaisintawee.

**Funding acquisition:** Daniel Boudreaux.

**Investigation:** Thunyarat Anothaisintawee, Krit Harncharoenkul, Kamonporn Poramathikul.

**Methodology:** Thunyarat Anothaisintawee, Kittijarankon Phontham, Michele Spring, Daniel Boudreaux, Jeffrey Livezey, Narisara Chantratita.

**Project administration:** Thunyarat Anothaisintawee.

**Resources:** Worachet Kuntawunginn.

**Software:** Thunyarat Anothaisintawee.

**Supervision:** Michele Spring, Daniel Boudreaux, Jeffrey Livezey, Narisara Chantratita.

**Validation:** Thunyarat Anothaisintawee, Krit Harncharoenkul, Kamonporn Poramathikul, Parat Boonyarangka, Narisara Chantratita.

**Writing – original draft:** Thunyarat Anothaisintawee.

**Writing – review & editing:** Krit Harncharoenkul, Kamonporn Poramathikul, Kittijarankon Phontham, Parat Boonyarangka, Worachet Kuntawunginn, Michele Spring, Daniel Boudreaux, Jeffrey Livezey, Narisara Chantratita.

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
