## [Decision Letter · Decision Letter 0]

13 Feb 2023

Dear Dr. Anothaisintawee,

Thank you very much for submitting your manuscript "Full tile: Efficacy of drug treatment for severe melioidosis and eradication treatment of melioidosis: A systematic review and network meta-analysis" for consideration at PLOS Neglected Tropical Diseases. As with all papers reviewed by the journal, your manuscript was reviewed by members of the editorial board and by several independent reviewers. In light of the reviews (below this email), we would like to invite the resubmission of a significantly-revised version that takes into account the reviewers' comments. 

We cannot make any decision about publication until we have seen the revised manuscript and your response to the reviewers' comments. Your revised manuscript is also likely to be sent to reviewers for further evaluation.

Sincerely,

Husain Poonawala

Academic Editor

Ana LTO Nascimento

Section Editor

Reviewer's Responses to Questions

**Key Review Criteria Required for Acceptance?**

**Methods**

-Are the objectives of the study clearly articulated with a clear testable hypothesis stated?

-Is the study design appropriate to address the stated objectives?

-Is the population clearly described and appropriate for the hypothesis being tested?

-Is the sample size sufficient to ensure adequate power to address the hypothesis being tested?

-Were correct statistical analysis used to support conclusions?

-Are there concerns about ethical or regulatory requirements being met?

Reviewer #1: -

Reviewer #2: Yes but see my comments below

Reviewer #3: I can not comment in detail of the methods of the technique of NMA used in this submission. The authors seem to have followed PRISMA guidelines and acknowledge limitations in the methods.

**Results**

-Does the analysis presented match the analysis plan?

-Are the results clearly and completely presented?

-Are the figures (Tables, Images) of sufficient quality for clarity?

Reviewer #1: -

Reviewer #2: See comments beklow

Reviewer #3: I think results are presented reasonably and it is discussed that many of the results do not reach significance or are under-powered. Some of the results were focused a bit on treatment regimens that have little current relevance.

**Conclusions**

-Are the conclusions supported by the data presented?

-Are the limitations of analysis clearly described?

-Do the authors discuss how these data can be helpful to advance our understanding of the topic under study?

-Is public health relevance addressed?

Reviewer #1: The treatment of melioidosis remains a significant issue in endemic areas. This study provides valuable data from numerous landmark trials for clinicians to select the most suitable regimens for their patients. The thoroughness of the data is a strength of the study. However, there are a few concerns regarding the discussion section.

1. Each study included in the paper was conducted to address a specific question. For example, Anunnatsiri et al (2020) concluded that patients with melioidosis who had no residual foci of infection after 12 weeks of oral co-trimoxazole could safely discontinue antibiotics. Chetchotisakd et al (2014) found that adding doxycycline to co-trimoxazole did not provide any additional benefit for maintenance therapy compared to co-trimoxazole monotherapy. Pooling the data across studies may lead to a misleading conclusion for clinicians.

2. Each study was conducted at a specific time period (from 1989 to 2020). Over the course of more than 30 years, there have been significant improvements in medical care, particularly in supportive treatment in critical care units, which have played a major role in lowering mortality rates. The author asserts that granulocyte-colony stimulating factor (G-CSF) is the most efficacious treatment for severe melioidosis, but this may not be the case in present times. Due to the latest sepsis campaign, G-CSF is not recommended for the treatment of sepsis and septic shock. The latest randomized controlled trial of G-CSF for melioidosis was published in 2007, and it is unclear whether its benefit is still applicable.

Reviewer #2: See comments below

Reviewer #3: Mostly- but I think the limitations and context of having to use NMA for the analysis could have maybe been better contextualized or discussed in more detail to a general audience who may not be to familiar with the statistics and why the authors used these methods.

**Editorial and Data Presentation Modifications?**

Reviewer #1: -

Reviewer #2: Some minor editing of English required

Reviewer #3: Minor Revision.

**Summary and General Comments**

Reviewer #1: In my opinion, the author should avoid concluding which option is the most efficacious treatment and instead provide a narrative discussion, highlighting the two limitations mentioned above in the text.

Reviewer #2: This paper describes a network meta-analysis of treatments for melioidosis based on an analysis of 9 studies of intensive phase treatment and 6 of oral eradication treatment.

My major concern is that the conclusions, particularly those relating to intensive phase treatment, are presented in the summary in such a way as to have a potentially major impact on the way melioidosis is treated without sufficient supporting evidence. The statement “ceftazidime plus G-CSF was the most efficacious treatment for severe melioidosis” implies that anything else might represent sub-optimal treatment, whereas detailed reading of the paper makes it clear that the difference was not statistically significant and that “further multi-center RCTs with larger sample sizes are required to confirm the benefit of G-CSF in patients with severe melioidosis”. In addition, G-CSF may well not be practical or affordable (or indeed necessary) for routine use in many melioidosis-endemic areas. I would suggest that this statement should not be made anywhere in the paper without relevant caveats, such as the statements in the discussion that “further multi-center RCTs with larger sample sizes are required to confirm the benefit of G-CSF in patients with severe melioidosis” and “the benefit of G-CSF may be greater in patients with lower severity of disease”.

I am also concerned that the conclusion that ceftazidime plus TMP-SMX was the second most effective treatment for decreasing hospital mortality in patients with severe melioidosis might encourage those treating melioidosis patients to use TMP-SMX routinely, thereby exposing many patients to unnecessary risks of adverse effects from TMP-SMX. I think that the approach advocated by the Darwin group, whereby TMP-SMX is only added in a selected sub-group of patients (Sullivan RP, Marshall CS, Anstey NM, Ward L, Currie BJ. 2020 Review and revision of the 2015 Darwin melioidosis treatment guideline; paradigm drift not shift. PLoS neglected tropical diseases. 2020;14(9):e0008659), is more appropriate.

Secondly, whilst I recognise that the methodology of this sort of analysis requires that only RCTs can be considered in the meta-analysis, this automatically means that none of the extensive experience from Australia, where the number of cases of melioidosis seen is insufficient to enable RCTs to be undertaken, can be taken into account. In fact recent experience in Darwin has shown that their current guidelines (see above) have achieved a reduction in overall mortality to less than 10% (Currie BJ, Mayo M, Ward LM, Kaestli M, Meumann EM, Webb JR, et al. The Darwin Prospective Melioidosis Study: a 30-year prospective, observational investigation. The Lancet Infectious diseases. 2021;21(12):1737-46). Such guidelines are able to take into account the considerable variations between patients in the sites and extent of infection which it would be almost impossible to do in an RCT. At the very least this might warrant mention in the discussion. This would also enable discussion of the judicious and selective use of carbapenems as discussed in this and other papers from Australia. It is unfortunate that an RCT comparing ceftazidime and meropenem was started in Thailand many years ago (see https://clinicaltrials.gov/ct2/show/NCT00579956) but never finished due to the withdrawal of support by the pharmaceutical company, and this too may be worthy of mention.

It might also be worth mentioning in the introduction that the last Cochrane review on melioidosis was conducted in 2002 (https://www.cochranelibrary.com/cdsr/doi/10.1002/14651858.CD001263/full?highlightAbstract=melioidosi%7Cmelioidosis).

The finding that TMP-SMX for 20 weeks had the lowest probability of drug discontinuation, even when compared with TMP-SMX for 12 weeks, is counter-intuitive and warrants some discussion.

Minor Comments

Line 50. Ceftazidime misspelled.

Line 51. See my comment above about mortality, which over the past 5 years has been only 6%.

Line 52. Change ‘factors’ to ‘factor’ and add ‘or’ before trimethoprim-sulfamethoxazole.

Line 53. Change ‘mono-antibiotics’ to ‘antibiotic monotherapy’.

Line 71. These statements should be referenced. I suspect that the authors are using the 2004 version of the WHO Laboratory Biosafety Manual (3rd edition) which has now been superseded.

Lines 75-76. Reference 5 has actually been retracted as it contained some grossly misleading material (see https://pubmed.ncbi.nlm.nih.gov/32609720/). Furthermore, it is actually very misleading to quote a single mortality rate (21%) as this varies hugely in place and time - the most recent estimate in Australia is less than 10% - see my comments above.

Lines 89-90. Again, it is very misleading to quote a single value (23%) for recurrence rates as it is hugely variable and dependent on a number of factors.

Lines 92-4. This statement requires a reference.

Line 282. Add ‘with’ after ‘compared’.

Line 367. Review punctuation in this sentence.

Reviewer #3: Efficacy of drug treatment for severe melioidosis and eradication treatment of melioidosis: A systematic review and network meta-analysis.

Reviewer overview:

This is an interesting analysis of drug therapy for melioidosis where perhaps good quality data is lacking. It is generally well written. The use of a “A two stage NMA with surface under the cumulative ranking curve (SUCRA) was used to estimate the comparative efficacy of treatment regimens” is interesting and may be unfamiliar to many readers, including myself as reviewer. 

It seems there is some inference with this method, especially were no direct comparison is made in the included studies. 

If methods employed are sound (as assessed by someone knowledgeable in NMA) then I think this study has utility of suggesting further assessment of using adjunct G-CSF in severe melioidosis therapy and further well-planned prospective studies to assess the best eradication therapy. 

I have no major concerns (assuming the analysis is performed correctly and is a valid application) but please see general comments, along with minor comments for revision. 

General comments: 

All the studies were from Thailand (line 246), this seems a bit unusual given your search strategy did not specify studies had to be a certain geographic area? Any comments on this and how it might affect your results/ implications/ conclusions? Given specific treatment modalities / local Thai strains or other factors. 

What are the current antibiotic guidelines for treating melioidosis in Thailand and do these differ from other country guidelines? Is G-CSF recommended in any Thailand treatment guidelines for melioidosis? 

Much of the discussion seems to focus a lot on comparison to Chloramphenicol/Doxy/TMP-STX to CAZ + G-CSF, however this seems a bit misplaced as this “triple therapy” should be rarely used now and is not really standard of care – I think the focus should be more on is CAZ + G-CSF better then monotherapy with CAZ or a carbapenem. In general, as a clinician many of the comparison are not overly useful. 

Is it worth outlining a bit more what SUCRA values actually mean to people who are not familiar with NMA? Its easier for most readers to understand what RR are whether this is significant or not – however much of your results rely on the SUCRA values, when for example CAZ vs CAZ + G-CSF is not significant, RR 0.81 (0.61-1.06). But then your SUCRA for CAZ + G-CSF is 79.7% (the highest) and therefore the conclusion is this is most likely the best therapy. I think it could be discussed contextualised a bit better how you have come to this conclusion. 

Minor:

Line 31: Full stop/ period after regimens. 

Line 50: Many guidelines suggest meropenem, any reason to state imipenem instead? 

Line 70: Small distinction, but it is the U.S. Centers [with an s] of Disease…

Line 74: Suggest delete “continuously” 

Line 76: “But the death rate doubly increases when patients have septicemia”- rewrite without the phase “doubly increases” Note also ref 5 has had a retraction due to significant issues with the article – consider not using this reference. An issue I noted, for example, was was that the 21% mortality rate stated (in Thailand) had a reference to a case report of glanders in a human, or is not referenced when this same rate is stated to be the mortality rate in Australia. 

Line 77: Carbapenem is not an antibiotic as such, rather a class of antibiotics 

Line 78-79: “Mortality rate was still high (37%), especially in patients with septicemia” This might need a bit more context as you state antibiotics reduced mortality rate but in a previous sentence say the mortality rate is approx 41% with reference to another study. Is this comparing untreated/ undertreated to optimally treated etc? 

Line 83-84: “Unfortunately, none of these studies demonstrated significant benefit of these treatment regimens over ceftazidime to reduce mortality in severe melioidosis patients”- were all these studies comparing a treatment regimen to ceftazidime? 

Line 132-134: Often the reviewers who did the review for selection and the 3rd party consensus would have this defined by bracketed initials. 

Line 154: What do you mean by “clinical-confirmed” recurrence?

Line 155: Typo- Clulture

Line 164: As above which reviewer initials? 

Line 164-166: “Outcomes of interest including frequency of patients who developed or did not develop the outcomes for each treatment were also extracted”- rewrite. 

Line 176-178: As per prior comments consider specifying which reviewers did what. 

Line 201-202 and Fig 1, line 229-235: It is a bit confusing from reading the text and comparing to this figure- did you have 8 or 9 studies that underwent analysis for treatment of severe melioidosis? 

Line 247-249: Make it clear you are also not just looking at a catch all “co-morbid conditions” “The rates of the co-morbid conditions of diabetes mellitus and chronic kidney disease ranged..”

Line 250: Keep consistent decimal points. 

Line 260, Tab 1 Sookpranee, 1992: Pneumonia box is empty. 

Line 267-273: “Treating with chloramphenicol plus doxycycline plus TMP-SMX had a significantly greater risk of hospital mortality than other treatments except for cefoperazone-salbactam plus TMP-SMX, with RRs of 2.22 (95% CI: 1.28-3.86) for ceftazidime plus TMP-SMX, 1.96 (95% CI: 1.13-3.38) for amoxicillin-clavulanic acid, 2.01 (95% CI: 1.12-3.59) for Imipenem, 2.38 (95% CI:1.39-4.07) for ceftazidime plus G-CSF, and 1.92 (95% CI: 1.21-3.05) for ceftazidime. Ceftazidime plus G-CSF had a lower hospital mortality rate than all other treatments, although the results were not statistically significant” 

Where did you get the RR of 2.22 from- this is not in the table? 

I take it you single out the treatment regimen of chloramphenicol/ doxycycline/ TMP-SMX because this had the most significant results compared to other treatments but is this clinically relevant – is this regimen used much in practice? It might be worth highlighting some of the RR, even if not significant for other treatment regimens that are often used. 

Line 280-297, Tab 2: A somewhat busy and hard to interpret table. This might be my unfamiliarity with NMA, but clearly comparisons are made between treatment regimens such as imipenem vs co-amoxyclav or TMP-STX + doxy + chloramphenicol vs imipenem (which is a significant result). Is might be worth outlining which results are “derived” or indirect vs from direct comparison from studies (which would seem to hold more weight). 

Line 293: “disseminated disease”- how is this defined? Compared to bacteraemia. 

Line 318-323, Tab 4: Similar to comments on table 2. Consider outlining direct comparison vs NMA derived values. 

Line 371-376 Tab 5, comments per other tables. 

Line 403-404: “Currently, ceftazidime with or without TMP-SMX is the preferred first line treatment for severe melioidosis”- reference any relevant guidelines. Define specifically when addition of TMP-SMX is thought to be of benefit – i.e. patients with focal disease rather than just pneumonia/ bacteraemia. 

Line 478-479: “Inclusion of more data from more studies would strengthen our conclusion”. Well, it might not – I think you need to highlight that your findings are perhaps more hypothesis generating, far from good quality evidence to suggest G-CSF should definitively be utilised and that we need well designed prospective studies looking into therapy for melioidosis. I think the fact that direct meta-analysis could not be performed on “standard therapies” for melioidosis and NMA had to be performed – with many results lacking power / not reaching significance- really highlights the need for good quality clinical trials to improve treatment of melioidosis.

PLOS authors have the option to publish the peer review history of their article (what does this mean?). If published, this will include your full peer review and any attached files.

Reviewer #1: No

Reviewer #2: No

Reviewer #3: No
---

## [Decision Letter · Decision Letter 1]

19 Apr 2023

Dear Dr. Anothaisintawee,

Thank you very much for submitting your manuscript "Full tile: Efficacy of drug treatment for severe melioidosis and eradication treatment of melioidosis: A systematic review and network meta-analysis" for consideration at PLOS Neglected Tropical Diseases. As with all papers reviewed by the journal, your manuscript was reviewed by members of the editorial board and by several independent reviewers. The reviewers appreciated the attention to an important topic. Based on the reviews, we are likely to accept this manuscript for publication, providing that you modify the manuscript according to the review recommendations. 

The editor respectfully requests that the manuscript be reviewed from clarity and grammar before it can be accepted for publication.

Sincerely,

Husain Poonawala

Academic Editor

Ana LTO Nascimento

Section Editor

The editor respectfully requests that the manuscript be reviewed from clarity and grammar before it can be accepted for publication.

Reviewer's Responses to Questions

**Key Review Criteria Required for Acceptance?**

**Methods**

-Are the objectives of the study clearly articulated with a clear testable hypothesis stated?

-Is the study design appropriate to address the stated objectives?

-Is the population clearly described and appropriate for the hypothesis being tested?

-Is the sample size sufficient to ensure adequate power to address the hypothesis being tested?

-Were correct statistical analysis used to support conclusions?

-Are there concerns about ethical or regulatory requirements being met?

Reviewer #1: The author addressed a clear objective in the study, and the design used to prove the hypothesis could be considered appropriate.

Reviewer #2: (No Response)

Reviewer #3: -

**Results**

-Does the analysis presented match the analysis plan?

-Are the results clearly and completely presented?

-Are the figures (Tables, Images) of sufficient quality for clarity?

Reviewer #1: The study results were completely presented, and the figures were easy to understand.

Reviewer #2: (No Response)

Reviewer #3: -

**Conclusions**

-Are the conclusions supported by the data presented?

-Are the limitations of analysis clearly described?

-Do the authors discuss how these data can be helpful to advance our understanding of the topic under study?

-Is public health relevance addressed?

Reviewer #1: After revision, the conclusions are clearer, and the chance of misleading the reader is less than in the previous version.

Reviewer #2: (No Response)

Reviewer #3: -

**Editorial and Data Presentation Modifications?**

Reviewer #1: The author completely addressed my concern, and I have no further corrections.

Reviewer #2: (No Response)

Reviewer #3: Needs just minor revision of grammar / spelling of edited sections. Also needs consistent spacing of references - sometimes references[1] are immediately after and sometimes spaced [1] as demonstrated.

**Summary and General Comments**

Reviewer #1: The treatment of melioidosis remains a significant issue in endemic areas. This 

study provides valuable data from numerous landmark trials for clinicians to select the most 

suitable regimens for their patients. The thoroughness of the data is a strength of the study.

As for my concerns from previous versions, applying network meta-analysis (NMA) may not lead to a conclusive answer to the author's question. Therefore, readers must carefully consider their interpretation.

Reviewer #2: I am reasonably happy that the authors have now included sufficient caveats to their conclusions to address my original concerns. However, despite the assurance that the English has been reviewed by a native speaker I did find a number of instances where I felt it could be improved, particularly in the new text, much of which reads very much as though it has been written by a non-native and some of which is a little difficult to understand. I have included a few specific points below but I would suggest that the whole paper should be reviewed again by the native speaker, paying particular attention to the new text.

Additionally, altough the authors have attempted to address my comment about the reference to the WHO Biosafety Manual, they now inapproporiately suggest that the 2020 (4th) edition of the WHO Laboratory Biosafety Manual suggests that B. pseudomallei is "considered a risk group 3 pathogen by the World Health Organization (WHO)". In fact this edition goes to some lengths to say:

"Regardless of the approach used, the classification of biological agents and/or the

work being performed with them should not be considered static, nor should it be

universally applied across jurisdictions. Classification can vary according to contextual

factors (for example, geography, time, process), so the application of one country’s

classification system to another country should be avoided as it could create confusion

and result in inadequate or excessive risk control measures". 

Finally, reference 8 persists - it has been withdrawn and should not be cited.

Examples of English issues (lines cited are from the revised Word version)

L 91. Add 'the' before 'mortality rate'.

L 185. Recurrence misspelled. In addition I would suggest using 'clinically suspected' rather than 'clinical confirmed'.

L 225 and elsewhere. The spaces between the text and the references have been lost.

LL 283 and 286. Use 'studies'' (plural) not 'study's' (singular).

L 311-2. I would say 'failed to achieve staistical significance'.

L 312. Suggest replacing 'reduced' with 'lower'.

L 448-9. Rewrite.

L 473-5. Rewrite.

L 496-509. Review and rewrite.

L 522-529. Review and rewrite.

L 544-553. Review and rewrite.

L 558. Rewrite

L 562. Explain what you mean.

Reviewer #3: I think the authors have done a good job of responding to reviewer comments.

PLOS authors have the option to publish the peer review history of their article (what does this mean?). If published, this will include your full peer review and any attached files.

Reviewer #1: No

Reviewer #2: Yes: David AB Dance

Reviewer #3: No

Figure Files:

Data Requirements:

Reproducibility:

References

---

## [Editor Report · Decision Letter 2]

16 May 2023

Dear Dr. Anothaisintawee,

We are pleased to inform you that your manuscript 'Full tile: Efficacy of drug treatment for severe melioidosis and eradication treatment of melioidosis: A systematic review and network meta-analysis' has been provisionally accepted for publication in PLOS Neglected Tropical Diseases.

Best regards,

Husain Poonawala

Academic Editor

Ana LTO Nascimento

Section Editor

---

## [Editor Report · Acceptance letter]

7 Jun 2023

Dear Dr. Anothaisintawee,

We are delighted to inform you that your manuscript, "Efficacy of drug treatment for severe melioidosis and eradication treatment of melioidosis: A systematic review and network meta-analysis," has been formally accepted for publication in PLOS Neglected Tropical Diseases.

Best regards,

Shaden Kamhawi

co-Editor-in-Chief

Paul Brindley

co-Editor-in-Chief
